# Expression of ZNF695 Transcript Variants in Childhood B-Cell Acute Lymphoblastic Leukemia

**DOI:** 10.3390/genes10090716

**Published:** 2019-09-16

**Authors:** Ricardo De la Rosa, Vanessa Villegas-Ruíz, Marcela Concepción Caballero-Palacios, Eleazar Israel Pérez-López, Chiharu Murata, Martha Zapata-Tarres, Rocio Cárdenas-Cardos, Rogelio Paredes-Aguilera, Roberto Rivera-Luna, Sergio Juárez-Méndez

**Affiliations:** 1Experimental Oncology Laboratory, Research Department, National Institute of Pediatrics, Mexico City 04530, Mexico; Ricardo.dela.R.V@outlook.com (R.D.l.R.); vanessavillegasruiz@yahoo.com.mx (V.V.-R.); doc.marce1602@gmail.com (M.C.C.-P.); eleazarisraelp@yahoo.com.mx (E.I.P.-L.); 2Research Methodology Department, National Institute of Pediatrics, Mexico City 04530, Mexico; chiharumurata@gmail.com; 3Department of Pediatric Oncology, National Institute of Pediatrics, Mexico City 04530, Mexico; mzapatatarres@gmail.com (M.Z.-T.); oncoped_inp@hotmail.com (R.C.-C.); 4Department of Hematology, National Institute of Pediatrics, Mexico City 04530, Mexico; rapa3852@yahoo.com; 5Division of Pediatric Hem/Oncology, National Institute of Pediatrics, Mexico City 04530, Mexico; riveraluna@yahoo.com

**Keywords:** ZNF695, lncRNAs, leukemia, gene expression

## Abstract

B-cell acute lymphoblastic leukemia is the most commonly diagnosed childhood malignancy worldwide; more than 50% of these cases are diagnosed in Mexico. Although the five-year survival rate is >80%, 30% of patients experience relapse with poor prognosis. Cancer-associated gene expression profiles have been identified in several malignancies, and some transcripts have been used to predict disease prognosis. The human transcriptome is incompletely elucidated; moreover, more than 80% of transcripts can be processed via alternative splicing (AS), which increases transcript and protein diversity. The human transcriptome is divided; coding RNA accounts for ~2%, and the remaining 98% is noncoding RNA. Noncoding RNA can undergo AS, promoting the diversity of noncoding transcripts. We designed specific primers to amplify previously reported alternative transcript variants of ZNF695 and showed that six ZNF695 transcript variants are co-expressed in cancer cell lines. The amplicons were sequenced and identified. Additionally, we analyzed the expression of these six transcript variants in bone marrow from B-cell acute lymphoblastic leukemia patients and observed that ZNF695 transcript variants one and three were the predominant variants expressed in leukemia. Moreover, our results showed the co-expression of coding and long noncoding RNA. Finally, we observed that long noncoding RNA ZNF695 expression predicted survival rates.

## 1. Introduction

B-cell acute lymphoblastic leukemia (B-ALL) is the most commonly diagnosed childhood malignancy. Worldwide, B-ALL accounts for 85% of ALL cases [1]. The five-year survival rate is >90%, and the cure rate is ~80% in developed countries [2]. However, the cure rates are very low in developing countries [3]. Several factors are associated with leukemia, including risks associated with clinical classification, cytogenetic alterations, and delayed diagnosis. Genetic alterations are also associated with ALL; however, the association between ALL and molecular alterations such as differential gene expression, transcriptional regulation, and epigenetic modifications is unknown. Therefore, identifying molecules that drive malignancy is a current challenge to improve the diagnosis, prognosis, treatment and understanding of leukemia biology.

Transcriptional regulation in eukaryotes is orchestrated by a wide variety of molecules, including DNA, RNA and proteins. Many of these proteins are involved in DNA binding and promote the regulation of gene expression. The zinc finger (ZNF) protein family is the largest family of DNA-binding proteins in mammals. The ZNF proteins have a large number of motifs that include Cys2-His2, GATA, RanBP, A20, LIM, MYND, RING, PHD, and TAZ. Of these, the most common motifs are the Cys_2_-His_2_ domain-containing ZNF proteins [4,5]. On the other hand, another conserved domain present in one-third of all ZNF proteins [6] is Krüppel-associated box (KRAB) [7]. The coding sequences of approximately 50% of ZNF proteins with a KRAB domain (ZNF-KRAB) are located on chromosome 19q13 [8]. However, this cytogenetic location is not exclusive to ZNF-KRAB proteins. ZNF-KRAB proteins play several roles, including regulating gene expression mediated by RNA polymerases (Pol I, II, and III) [9], binding to transcriptional repressors [10,11,12,13,14,15,16], and regulating splicing [17,18,19,20,21,22,23,24]. The KRAB domain is transcribed by independent exons, an advantage for its nascent transcript, which can undergo alternative splicing (AS) [11,25,26,27,28], thus increasing the diversity of transcripts and the resulting proteins. However, the diversity of mRNAs encoding ZNF-KRAB proteins and the expression of these mRNAs in health or disease status are unknown.

In cell biology, many forms of processing occur by AS, including alternative 5′ splicing, alternative 3′ splicing, exon skipping, intron retention, mutually exclusive exon selection, and exon scrambling [29]. AS is regulated by specific sequences, including sites delimited by specific sequences called intronic definition (ID) and exonic definition (ED) elements [30] and a regulatory system comprising serine/arginine (SR)-rich proteins [30] and heterogeneous nuclear ribonucleoprotein particle (hnRNP) A/B proteins [31]. Specific pre-mRNA sequences play an important role as enhancers and silencers and can be classified as exonic splicing enhancers/silencers (ESEs/ESSs) or intronic splicing enhancers/silencers (ISEs/ISSs) according to their locations [32]. Diverse reports have shown that alterations in ID elements [30,33,34,35,36,37] as well as mutations in ESEs, ESSs, ISEs, and ISSs can promote intron/exon retention, skipping [34], or both.

Aberrant AS contributing to structural protein variations results in functional and nonfunctional end products [38], and aberrant AS has been associated with diverse cancer-associated processes, including cell death resistance, angiogenesis induction, genomic instability, tumor development promotion [38], and cancer progression [39,40]. The contribution of aberrant protein expression to the diversity of the cancer proteome and the functional impact of these proteins is unclear, and the diversity of alternative transcripts expressed is unknown. However, 98% of the human transcriptome is noncoding RNA. Noncoding RNA is divided into short noncoding RNA and long noncoding RNA (lncRNA). lncRNAs are transcripts with a full length of greater than >200 nucleotides [41]. These transcripts are classified based on their location and include intergenic, intronic, intronic antisense, overlapping, and overlapping antisense transcripts [42]. The expression of lncRNAs is highly spatially and temporally restricted [43]; some cellular compartments can be enriched in specific lncRNAs. Moreover, lncRNAs can exhibit very diverse activities, acting in the cytoplasm and the nucleus as chromatin regulators, RNA-binding proteins, promoters, or enhancers [44,45]. However, the diversity and functions of lncRNA expression are unclear.

Using high-density microarrays, we previously employed ovarian normal and tumor tissues to identify AS of ZNF695. Our findings showed two new alternative mRNA splice variants associated with ovarian cancer [46]. However, the full-length sequence was not evaluated. In this work, we employed HeLa, MCF7, RS4 and SUP-B15 cells to evaluate the full-length transcript variants of ZNF695. Our findings showed the coexpression of six alternative transcripts of the ZNF695 gene. Additionally, we identified the prevalence of ZNF695 transcript variant 3 (a lncRNA) in B-ALL patients and the ability of this variant to predict overall survival.

## 2. Materials and Methods

### 2.1. Cell Lines and Growth Conditions

Cells were cultured at 37 °C and 5% CO_2_ in their preferred medium supplemented with 10% fetal bovine serum and 1% penicillin-streptomycin. The following culture media were used: HeLa cells were cultured in Dulbecco’s modified Eagle’s medium (DMEM, Gibco, Life Technologies, Carlsbad, CA, USA); MCF-7 and RS4 cells were cultured in RPMI-1640 medium (Sigma-Aldrich, St. Louis, MO, USA); and SUP-B15 cells were cultured in Iscove’s modified Dulbecco’s medium.

### 2.2. Patients and Ethics Statements

Bone marrow samples were obtained from patients with a diagnosis of ALL who had previously provided signed informed consent, and the protocol was approved by the Institutional Ethics Committee (INP protocol 060/2016) and was in accordance with the Declaration of Helsinki. Leukemia cells were isolated using Lymphoprep density gradient medium (STEMCELL Technologies, St. Kent, WA, USA). According to the protocol, PBS was added to bone marrow (1:1, *v/v*), and the mixture was transferred to 3 mL of Lymphoprep and centrifuged at 1500 rpm for 30 min. Leukemia cells retained in the interface were transferred to a new tube and diluted with PBS (1:1, *v/v*). The cell suspension was gently homogenized by inversion and centrifuged at 3000 rpm for 5 min. Recovered cells were stored at −70 °C until they were used for nucleic acid purification.

### 2.3. RNA Purification and Reverse Transcription

Total RNA was purified from cultured cell lines and leukemia cells. Briefly, for cell lines, 1 mL of TRIzol reagent (Ambion, Life Technologies, Carlsbad, CA, USA) was added to the culture dishes, and the cells were scraped and collected in a 1.5 mL tube. Cells were disrupted with a TissueLyser at a frequency of 25/s for 30 s, and RNA purification was performed following the TRIzol manufacturer’s recommendations. Finally, RNA was quantified using a NanoDrop One UV-Vis Spectrophotometer (Thermo Fisher Scientific, Waltham, MA, USA). For cDNA synthesis, 1 µg of total RNA was digested with DNase and incubated at 37 °C for 30 min, and 1 µL of 5 mM EDTA was then added to stop reaction the at 65 °C for 10 min. The cDNA synthesis reaction contained the following components: 1X RT buffer, 10 U Transcriptor Reverse Transcriptase (Sigma-Aldrich), 0.4 µM random primers, 1 mM dNTPs, and 20 U RNaseOut (Thermo Fisher Scientific). The reaction mixture was incubated for 10 min at 25 °C, for 30 min at 55 °C, and for 5 min at 85 °C.

### 2.4. Rapid Amplification of cDNA 3′ Ends (3′ RACE)

The 3′ RACE procedure was performed using a 3′ RACE System for Rapid Amplification of cDNA Ends Kit (Thermo Fisher Scientific) according to the protocol. cDNA synthesis was performed using 5 µg of total RNA. RNA was subjected to DNase treatment as previously described. After adding 10 µM adapter primer (AP), the mixture was placed in a thermal cycler for 10 min at 70 °C and was then transferred to ice for one minute. Finally, this mixture was added to a mixture containing 1X PCR buffer, 25 mM MgCl_2_, 50 mM dNTP mix, and a final concentration of 0.5 M DTT, and the mixture was placed in a thermal cycler for 5 min at 42 °C. Then, 200 U of SuperScript™ II Reverse Transcriptase was added, and the mixture was incubated for 50 min at 42 °C and 15 min at 70 °C and then placed on ice for one minute. Finally, 2 U of RNase H was added, and the mixture was placed in the thermal cycler for 20 min at 37 °C (Thermo Fisher Scientific).

### 2.5. PCR and Sequencing

PCR was performed using 25 ng of synthesized cDNA, and the reaction mixture contained 0.14 U Fast HotStart DNA Polymerase (KAPA2G, Kapa Biosystems, Wilmington, DE, USA), 0.2 mM dNTP mix, 1.5 mM MgCl_2_, 5 µM forward primer, 5 µM reverse primer, and nuclease-free water up to 10 µL. The reaction mixture was incubated for 1 min at 95 °C, 15 s at 95 °C, 15 s at T_m_ (the melting temperature of each primer is shown in Table 1), 15 s at 72 °C, and 7 min at 72 °C for the final extension.

PCR products were purified using a Zymoclean™ Gel DNA Recovery Kit (ZYMO Research, Irvine, CA, USA) according to established protocols. After that, PCR products were sequenced using a BigDye Terminator v3.1 Cycle Sequencing Kit (Applied Biosystems, Waltham, MA, USA), according to established protocols. The master mix was placed in a Proflex thermal cycler for 25 cycles: 30 s at 95 °C, 15 s at 50 °C and 4 min at 60 °C. The samples were sequenced using an Applied Biosystems ABI Prism 3130 Genetic Analyzer (Applied Biosystems). Finally, the sequences were analyzed using UGENE v1.23.1. The resulting sequences were aligned using Clustal Omega (Clustal Omega, EMBL-EBI, Cambridge, UK). The following reference sequences were used: ZNF695 transcript variant one (ZNF695_TV1, NM_020394.4), ZNF695 transcript variant two (ZNF695_TV2, NM_001204221.1), and ZNF695 transcript variant 3 (ZNF695_TV3, NR_037892.1, lncRNA).

### 2.6. Statistical Analysis

The clinical characteristics are summarized as absolute values and relative frequencies. Kaplan-Meier curves were generated to determine the survival and relapse rates. The differences in the survival rates among the ZNF695 transcript variants were determined by the Wilcoxon test. We considered *p* values of < 0.05 to indicate significant differences. Statistical analysis was performed using the commercial statistical package JMP11 from SAS Institute, Inc.

## 3. Results

### 3.1. Alternative ZNF695 Transcript Variants Are Expressed in Cancer Cell Lines

The ZNF695 gene is localized on Chromosome 1 and the reverse strand. Two transcripts encode proteins. The first, the longest transcript, consists of four exons with a total transcript length of 3341 bp. This variant is characterized by a very long exon 4 of 2933 bp. The variant (TV1) encodes a protein with 515 aa ZNF695-KRAB protein (Ensembl database ENST00000339986.8, NCBI database: NM_020394.5, TV1) that belongs to the ZNF and Cys2-His2 families. The second, the short transcript, has six exons with a length of 826 bp (ENST00000487338.6) and 919 bp in NCBI (NM_001204221, TV2). However, this protein contains no ZNF domain. Additionally, the ZNF695 gene is transcribed to the ZNF695 long noncoding RNA (ENST000000498046.2, 504 bp). However, in the NCBI database, the noncoding transcript has 923 bp (NR_037892.2, TV3), which has four nucleotides more than the transcript variant 2 NM_001204221. Finally, three nonsense-mediated decay transcripts are reported in Ensembl. These transcripts comprise six exon and show diverse sequences (ENST00000491337.6, ENST00000479214.5, ENST00000366504.6, with 714, 885 and 862 bp, respectively). Hereafter, we only used the sequences that are reported in the NCBI database.

The ZNF695 gene encodes the ZNF-KRAB protein based on its protein domains, suggesting a negative regulation. However, few studies have focused on the functions of spliced alternative mRNAs. Moreover, there are no studies that show the mechanism of AS of ZNF695. Previously, we identified the coexpression of three mRNA transcript variants of ZNF695 expressed in ovarian cancer, showing alternative 5’ splice sites in exon one and exon two [46]. Additionally, we previously identified the expression of ZNF695 variants in the Jurkat, FaDu, HEK-293, HEp2, MD-MB-231, CaSki, and HeLa cell lines [46]. These results suggest that the expression of ZNF695 is not specific to ovarian cancer. However, we did not elucidate the full-length transcript variants. To identify full-length new transcript variants of ZNF695, we employed cancer cell lines with different tissue origins (cervical cancer, breast cancer and leukemia). First, we amplified the RPL4 housekeeping gene in the HeLa, MCF-7, RS4, and SUP-B15 cell lines to confirm cDNA integrity (Figure 1A). After that, we corroborated the amplification of three previously reported ZNF695 transcript variants in the cancer cell lines and malignant ovarian tumors [46]. As expected, three amplicons of the 400 bp, 360 pb and 310 bp previously reported were found (Figure 1B). Then, we designed primers as shown in Table 1 to selectively amplify ZNF695_TV1 and ZNF695_TV2/TV3 (Figure 1C,D). Surprisingly, we observed expression in both PCR assays, showing that ZNF695_TV1 and ZNF695_TV2 or ZNF695_TV3 are co-expressed in cancer cell lines. ZNF695_TV1 showed differential expression among the cell lines, with SUP-B15 cells exhibiting the highest expression level. However, ZNF695_TV2/TV3 was expressed at low levels, mainly in HeLa and RS4 cells. Our findings suggest nine possible AS scenarios for ZNF695.

Then, we designed two forward primers that complemented SS1 and SS2, as well as two reverse primers—one specific to the 3′ end of ZNF695_TV1 (NM_020394.5), and the other specific to the 3′ end of ZNF695_TV2 (NM_001204221.1) or ZNF695_TV3 (NR_037892.1, a lncRNA). SS1 and ZNF695_TV1 primers amplified variants TV4 and TV5 at the same time, while SS2 and ZNF695_TV1 primers amplified specifically TV5. The second reverse primer was used to amplify TV6 when paired with SS1 forward primer, or TV7 variant when paired with SS2 primer (Table 1, Figure 2A–D).

The amplification of specific alternative transcripts showed that ZNF695_TV4 and ZNF695_TV5 were expressed in the four cell lines (Figure 2A,B). As expected and as indicated in Figure 2A, we obtained two PCR fragments because ZNF695_TV4 expression and ZNF695_TV5 expression are not mutually exclusive, and the ending sequence confirmed the identity, as shown in Figure 2G. ZNF695_TV6 and ZNF695_TV7 were expressed in only the MCF7 and HeLa cell lines (Figure 2C,D), and the ending sequence confirmed the identity as shown in Figure 2H. Interestingly, ZNF695_TV7 was expressed at low levels in both cell lines (Figure 2D). To determine the sequence of ZNF695_TV6 (Appendix A) and ZNF695_TV7 (Appendix A), we sequenced the PCR products. Surprisingly, the resulting sequences identified ZNF695_TV3 (NR_037892) (Figure 2E,F). From this result, we concluded that ZNF695_TV2 was not expressed in the cell lines analyzed. Additionally, we observed the co-expression of six ZNF695 transcript variants in these cancer cell lines.

### 3.2. Alternative 3′ Ends in ZNF695 Transcript Variants

To determine the full-length alternative transcripts of ZNF695, we performed 3′ RACE in the cell lines. First, we confirmed cDNA amplification with 3′ RACE via amplification of the RPL4 housekeeping gene (Figure 3A). Then, we performed PCR using three specific forward primer mixes (ZNF695_TV1, SS1 and SS2) and a universal amplification primer (UAP) as the reverse primer. With our design, we expected four amplicons: ZNF695_TV4, 3215 bp; ZNF695_TV5, 3049 bp; ZNF695_TV6, 781 bp; and ZNF695_TV7, 615 bp. The amplicons of ZNF695_TV6 and ZNF695_TV7 were of the expected size, as shown by the green and blue arrows, respectively (Figure 3B). However, the amplicons of ZNF695_TV4 and ZNF695_TV5 did not shown the expected size (Figure 3B). Additionally, the sequences of these amplicons could not be characterized because the sequences obtained were unreadable, suggesting unspecified PCR products. Thus, we did not obtain the full-length sequences of the ZNF695 transcript variants TV4, TV5, TV6, and TV7 using 3′ RACE. However, our previous sequences in Figure 2E–H showed that the ZNF695 gene could generate seven alternative transcripts, of which only six were expressed in the tested cancer cell lines, as shown in the AS model for ZNF695 in Figure 4.

### 3.3. Alternative ZNF695 Transcript Variants Are Expressed in B-ALL

Here, we identified six AS transcripts of ZNF695 mRNA in cancer cell lines, including a B-ALL cell line. To determine whether alternative ZNF695 transcripts are expressed in leukemia patients, mRNA from nine healthy donors and 43 B-ALL patients was analyzed. First, we amplified the housekeeping gene RPL4 from the cDNA of the healthy controls and patients (Figure 5A,B). The housekeeping gene was amplified in all samples as expected. We then amplified ZNF695_TV1 using specific primers according to Table 1. Interestingly, we observed weak expression in the healthy controls (Figure 5C). However, diverse expression patterns were observed in the patients, as shown in Figure 5D. Of the patients analyzed, 51.16% expressed ZNF695_TV1. Moreover, the expression of ZNF695_TV1 varied widely, as shown in Figure 5D and Figure 7. Next, we amplified ZNF695_TV3 with specific primers, as shown in Table 1 and Figure 6A. A total of 41.86% of the samples were positive, 10% fewer than for ZNF695_TV1. Moreover, we identified that ZNF695_TV3 expression and ZNF695_TV1 expression may be mutually exclusive because they were expressed in different samples as follows: ZNF695_TV3 (samples L4, L2, and L17) and ZNF695_TV1 (samples L14, L21, L36, among others) (Figure 7).

Then, we evaluated the expression of ZNF695_TV4, ZNF695_TV5, ZNF695_TV6, and ZNF695_TV7 in only samples positive for ZNF695_TV1 and ZNF695_TV3 expression (n = 26). We evaluated the expression of ZNF695_TV4, ZNF695_TV5, ZNF695_TV6, and ZNF695_TV7 in the positive samples because we previously observed no expression of ZNF695_TV4, ZNF695_TV5, ZNF695_TV6, or ZNF695_TV7 in the samples negative for ZNF695_TV1 and ZNF695_TV3 expression. We identified ZNF695_TV4, ZNF695_TV5, ZNF695_TV6, and ZNF695_TV7 expression in 16.2%, 37.2%, 6.9%, and 13.9% of these samples, respectively (Figure 6B–E and Figure 7). The leukemia cell lines did not express ZNF695_TV6 and ZNF695_TV7 (Figure 2C,D). However, the expression of these variants was found in some leukemia patients (Figure 6D,E). ZNF695_TV4 was expressed only slightly in the patients (Figure 6B) and cancer cell lines (Figure 2A).

Expression of the different ZNF695 transcript variants varied widely among the samples, and ZNF695_TV4 and ZNF695_TV6 had the lowest expression levels in the patients. We then calculated the relative expression via densitometric analysis in ImageJ according to established parameters, with the housekeeping gene RPL4 as the reference gene. Heterogeneity was found in the relative expression levels of the transcript variants as well as in the expression ratios among all samples analyzed (Figure 7). Great diversity was identified in the co-expression patterns of the ZNF695 transcript variants.

Finally, we estimated the overall survival of the patients via Kaplan-Meier curves based on ZNF695 transcript variant expression according to clinical characteristics Table 2. A significant association between overall survival and relapse and ZNF695_TV1 expression p = 0.0808 was not observed. However, the Kaplan-Meier curves based on ZNF695_TV3 expression revealed statistically significant differences in the overall survival and the tendency to relapse (Figure 8). These results showed that the expression of ZNF695_TV3 is associated with poor survival and increased tendency to relapse.

## 4. Discussion

AS plays an important role in cell biology. However, little is known about the diversity of the transcripts expressed in several tissues and human diseases, including cancer. Several AS transcripts contribute to protein diversity [47], but the diversity and function of RNA splice variants are unknown. In cancer, many AS transcripts encode proteins that contribute to the pathogenesis of the disease, conferring a gain, loss or change of function to the encoded protein. Tumor cells exhibit extensive dysregulation of normal biological processes. In these cells, AS is known as aberrant splicing, and this process is apparently a consequence of malignant transformation. Aberrant splicing occurs frequently in several types of cancers [48]. However, identifying AS events is very difficult because AS patterns vary widely and include events such as alternative 5′ and 3′ splicing, exon skipping, and intron retention.

The molecular diagnosis of cancer is a serious challenge to modern medicine. Identifying the molecular markers that can predict prognosis, in addition to assigning specific treatments to oncology patients such as ER/PR-positive and ER/PR-negative breast cancer patients, is difficult [49]. The genomic tools applied in cancer have revealed a comprehensive approach for detecting deregulation of the human transcriptome. We used high-density microarrays to identify new patterns of AS in ZNF695 transcripts [46] and performed a simple procedure to evaluate the expression of the ZNF695 transcript variants. We designed specific primers to evaluate the new isoforms identified in a previous report [46]. Surprisingly, we observed the co-expression of six alternative transcript variants in cancer cell lines and leukemia patients.

The complete sequence of the human transcriptome is unknown, but a gene:transcript ratio of 1:7 has been suggested [50]. However, the diversity of the human transcriptome in healthy and disease states is unclear. ZNF695 transcript variants showed very low expression in healthy ovarian tissue [46] or healthy lymphoid cells (Figure 5C). We believe that the expression of ZNF695 increases during carcinogenesis and the subsequent generation of aberrant alternative splice variants. Moreover, we found nine patterns of AS in association with the ZNF695 transcript variants expressed in leukemia patients (Figure 7).

We think that transcript variants one and three are regulated differentially because in some samples, the expression of these transcript variants is mutually exclusive. Thus, our results suggest two regulatory mechanisms for ZNF695 gene expression and AS. In contrast, ZNF695_TV4, ZNF695_TV5, ZNF695_TV6, and ZNF695_TV7 were not expressed alone (Figure 7). To date, no previous studies have identified and quantified the expression of novel ZNF695 transcript variants in childhood leukemia. Some hypotheses suggest that the diverse expression patterns result from the heterogeneity of tumor samples. We discovered four novel ZNF695 alternative transcripts that are co-expressed in cell lines and leukemia patients. The results of the sequence-based bioinformatic analysis for identify the initiation codons in the cDNA sequence using the ATGpr program available in http://atgpr.dbcls.jp [51] website suggested that ZNF695_TV1, ZNF695_TV4, and ZNF695_TV5 are coding RNAs, while ZNF695_TV3, ZNF695_TV6, and ZNF695_TV7 are lncRNAs. The diversity of these alternative transcripts of ZNF695 suggests differential functions, and changes in the resulting sequence are known to promote changes in the function of the resulting protein via topological changes, additionally recapitulating cancer-associated phenotypes such as angiogenesis promotion [52], proliferation [53], and apoptosis avoidance [54].

Aberrant expression of alternative splice variants is a common event in cancer and is likely generated from somatic mutations [55] or changes in the expression of the AS-associated proteins; however, this observation does not clarify the function of most of these resulting transcripts. The ZNF695 protein has been evaluated in breast cancer; interestingly, ZNF695 expression could classify the nonluminal A and luminal B subtypes [56]. We evaluated ZNF695 expression in B-ALL and found some expression patterns. No specific antibodies against the ZNF695 proteins resulting from AS have been developed. Moreover, the function of ZNF695 is unclear. However, methylation-mediated silencing confers a complete therapeutic response in primary esophageal squamous cell carcinoma tumors [57], suggesting that in normal cells, the ZNF695 gene is methylated and, consequently, unexpressed. Our results showed that some samples were positive (n = 26) and others were negative (n = 17) for ZNF695 expression; the negative samples were likely methylated. These variants are likely unmethylated in patients who express ZNF695 transcript variants, and the prognosis and survival of these patients are poor (Figure 8B). Indeed, Li C et al. observed a strong correlation between the mRNA expression of ZNF695 and adverse prognosis in adult ALL [58]. These results are very interesting and suggest that the expression of ZNF695 could be advantageous in malignances, probably via negative regulation of tumor suppressor genes.

The ZNF695 gene codes for three transcripts. In the full-length ZNF695 protein (ZNF695_TV1), which includes four exons, the terminal exon corresponds to the DNA-binding domain. However, the binding sites of the ZNF695_TV1 protein are unknown. The ZNF695_TV1 protein contains a KRAB-containing ZNF domain and belongs to a large family of proteins present in many species [12]. ZNF proteins that contain a KRAB domain function as transcriptional repressors [19]. Some KRAB proteins can bind to RNA and interact with RNA polymerase II [59]. We believe that the ZNF695_TV1 protein plays a role as a transcriptional repressor; however, we do not know the mechanism by which this repressive activity occurs. Moreover, we detected partial loss of the KRAB domain in ZNF695_TV4 and ZNF695_TV5, suggesting a change in the repressive function. However, the protein products of ZNF695 AS cannot be detected because there are no specific antibodies against these proteins.

Approximately 90% of the human genome transcribes noncoding RNAs [60,61], of which lncRNAs are a subcategory. Noncoding RNAs are classified based on the number of nucleotides and include short and long forms [62,63]. lncRNAs can be classified as intergenic, overlapping, sense, and antisense according to their location. We observed that ZNF695_TV3 is an overlapping noncoding RNA, and based on its size, it can be classified as a lncRNA. However, the function of this transcript is completely unknown. A significant relationship between ZNF695_TV3 expression and patient survival was identified (Figure 8A). The expression of the lncRNA ZNF695 transcript in cancer has not been reported. Moreover, several studies have suggested that lncRNAs regulate gene expression to mediate the interaction with chromatin and could play several roles, including oncogenic roles, as has been indicated for lncRNAs such as HOTAIR [64], MALAT1 [65], SPRY4IT1 [66], and H19 [67]. Our results are the first to show the expression of lncRNA ZNF695_TV3 and the role of this transcript as a predictor of survival in B-ALL patients. However, more extensive studies are necessary to identify the function of ZNF695_TV3.

## 5. Conclusions

The ZNF695 gene is transcribed as six alternative transcript variants in human cancer cell lines. These transcript variants were evaluated in B-ALL patients, and positive expression was found in 60.4% of the patients. We found that lncRNA ZNF695_TV3 expression was associated with poor survival and an increased tendency to relapse in patients with B-ALL. These findings are preliminary and require further validation in a large cohort.

## Figures and Tables

**Figure 1 genes-10-00716-f001:**
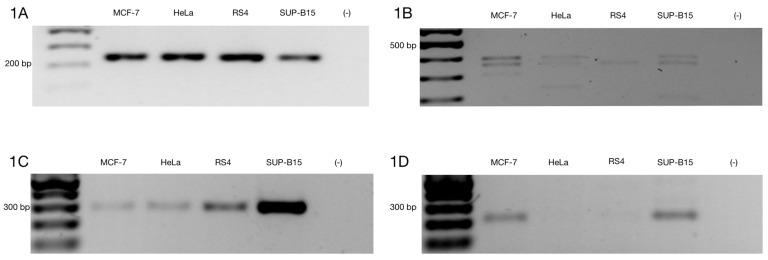
Alternative ZNF695 transcript variants are expressed in cancer cell lines. Gene expression was evaluated in MCF-7, HeLa, RS4, and SUP-B15 cell lines. (**A**) We observed homogenous expression of the RPL4 housekeeping gene in the four cell lines. (**B**) ZNF695 expression, including all transcript variants that were previously reported [46]. All cell lines expressed at least one of the ZNF695 transcripts, RS4 only expressed the ZNF695 variant TV1 and/or TV3 of ~400 bp, while the HeLa and SUP-B15 cell lines expressed two variants of ZNF695, which corresponded the expected lengths of 400 bp (TV1/TV3) and 360 bp (TV4/TV6). Additionally, we observed other variants of ~200 bp in these cell lines. However, we focused on the previous variants described. Finally, the MCF-7 cell lines co-expressed the three splicing variants of 400 bp (TV1/TV3), 360 bp (TV4/TV6) and 310 bp (TV5/TV7) [46]. (**C**) ZNF695_TV1 expression in the four cell lines. We designed specific primers for the 3′ end of the ZNF695 transcript variant 1 and a size of 279 bp (Table 1). The SUP-B15 cell line exhibited the highest expression. (**D**) We designed specific primers for the 3′ end of the ZNF695 transcript variant 2 or 3 and a size of 212 bp (Table 1). ZNF695_TV2/3 was expressed in two cell lines (MCF-7 and SUP-B15) and barely expressed in HeLa and RS4 cells.

**Figure 2 genes-10-00716-f002:**
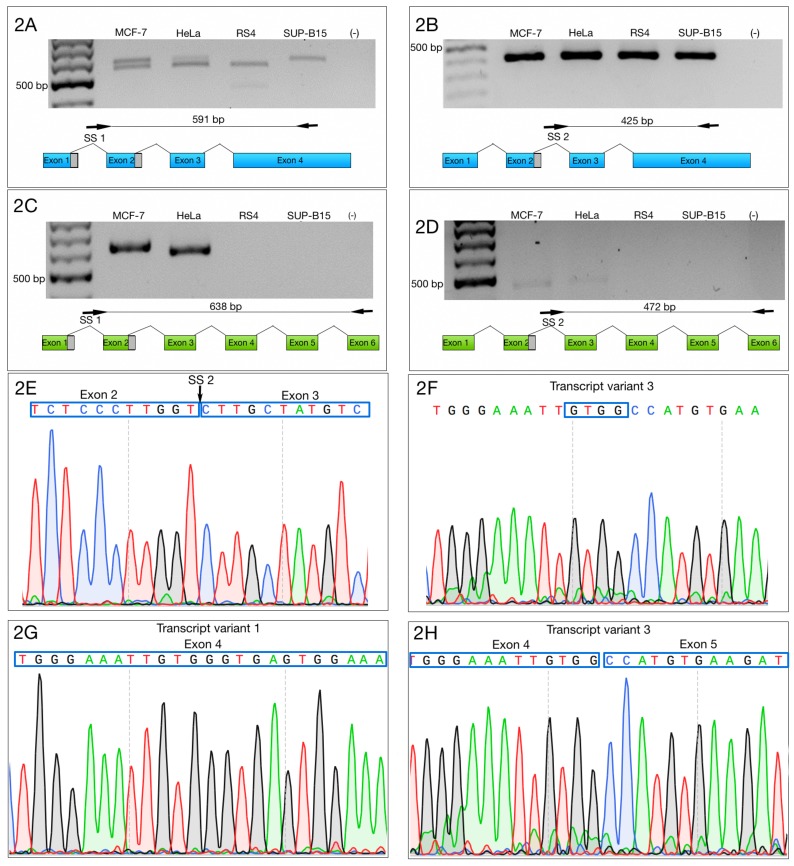
Selective amplification of ZNF695 alternative transcript variants. We selectively amplified the transcript variants of ZNF695. The black arrow and the black line indicate the sites of amplification and the PCR product size, respectively. The blue box indicates the full length of the long transcript of ZNF695 (TV1), the green box represents the short transcript of ZNF695 (TV3) and the grey box indicates the spliced region in the transcript. (**A**) Amplification of ZNF695_TV4 (591 bp) and ZNF695_TV5 (561 bp), which were co-expressed in the MCF-7 and HeLa cell lines. We observed that RS4 and SUP-B15 only expressed one variant, TV5 and TV4, respectively. (**B**) Specific amplification of ZNF695_TV5 of 425 bp in the four cell lines. (**C**) Amplification of ZNF695_TV6 (638 bp) in the MCF-7 and apparently HeLa cell lines expressed a preference for a TV7 of 608 pb. (**D**) Expression of ZNF695_TV7 (472 bp) was low in the MCF-7 and HeLa cell lines. We observed that in the HeLa cell line the amplified fragment showed bigger size than MCF-7. However, the sequences obtained from both fragments showed no difference. (**E**) Sequence of SS2 of the amplified ZNF695 transcript variants. The blue boxes show the boundary (exon 2–3) of the AS site. (**F**) Sequence that confirms the identity of ZNF695_TV3. (**G**) Partial sequence that confirms the identity of ZNF695_TV1 (NM_020394.5). (**H**) Partial sequence that confirms the identity of ZNF695_TV3 (NR_037892.1).

**Figure 3 genes-10-00716-f003:**
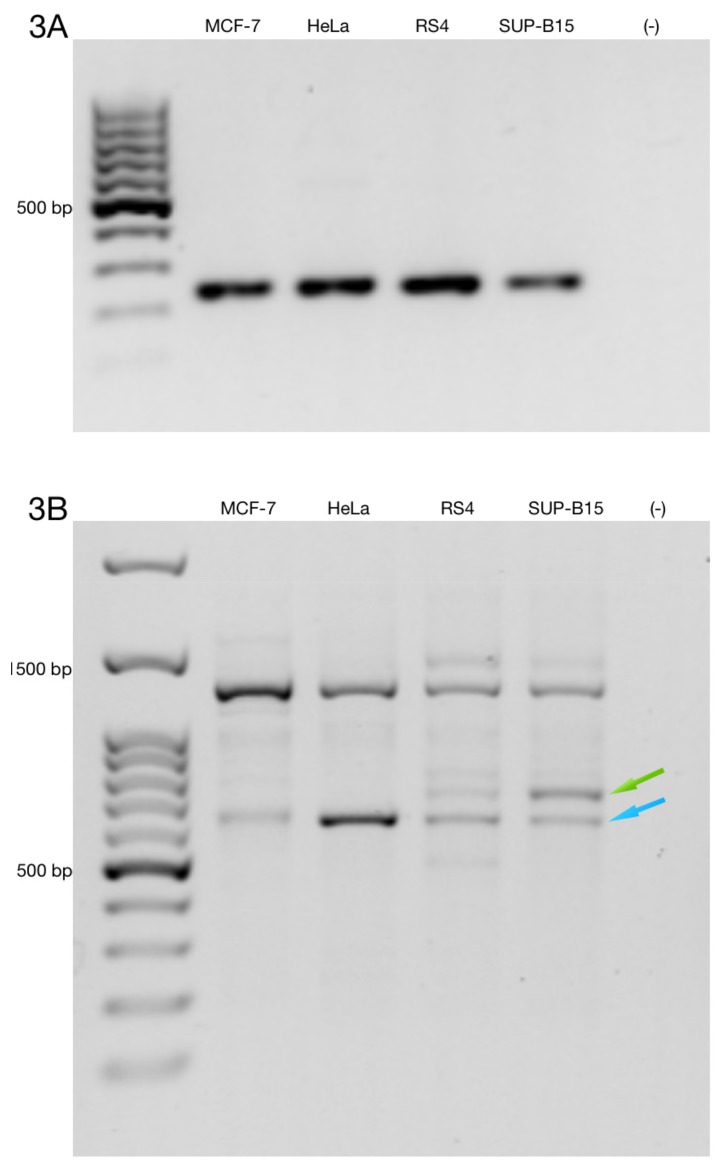
Rapid amplification of 3′ cDNA ends for the ZNF695 transcript variants expressed in the cancer cell lines. (**A**) Expression of the RPL4 housekeeping gene. (**B**) Co-expression of different amplicons of ZNF695. We observed two expected amplicons using the designed primers, as shown by the green and blue arrows; these amplicons correspond to ZNF695_TV6 and ZNF695_TV7, respectively.

**Figure 4 genes-10-00716-f004:**
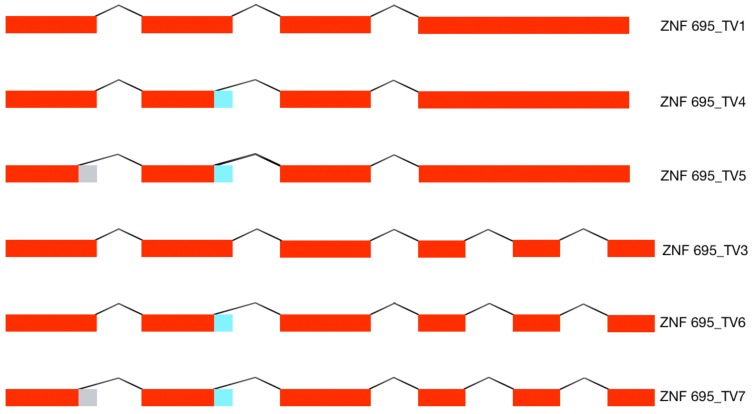
AS model of the ZNF695 transcript variants. The figure shows the six alternative transcripts expressed in the four cell lines. The red boxes indicate the coding sequences (exons), the gray boxes indicate the alternative splice site 1 (SS1), and the blue boxes indicate the alternative splice site 2 (SS2) identified in the cancer cell lines.

**Figure 5 genes-10-00716-f005:**
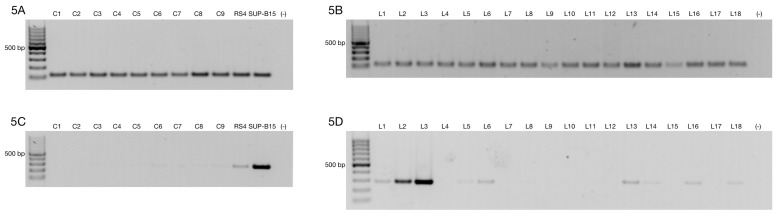
Comparison of ZNF695_TV1 expression in healthy subjects and B-ALL patients. (**A**) The RPL4 housekeeping gene was expressed in nine healthy donors and two positive controls (the RS4 and SUP-B15 cell lines). Agarose gel evaluating cDNA integrity via expression of the housekeeping gene. (**B**) Representative gel of the 43 B-ALL patients analyzed. The RPL4 housekeeping gene was expressed in all samples. (**C**) The agarose gel showed very low expression of ZNF695_TV1 in the healthy controls. (**D**) Agarose gel showing ZNF695_TV1 expression in B-ALL patients.

**Figure 6 genes-10-00716-f006:**
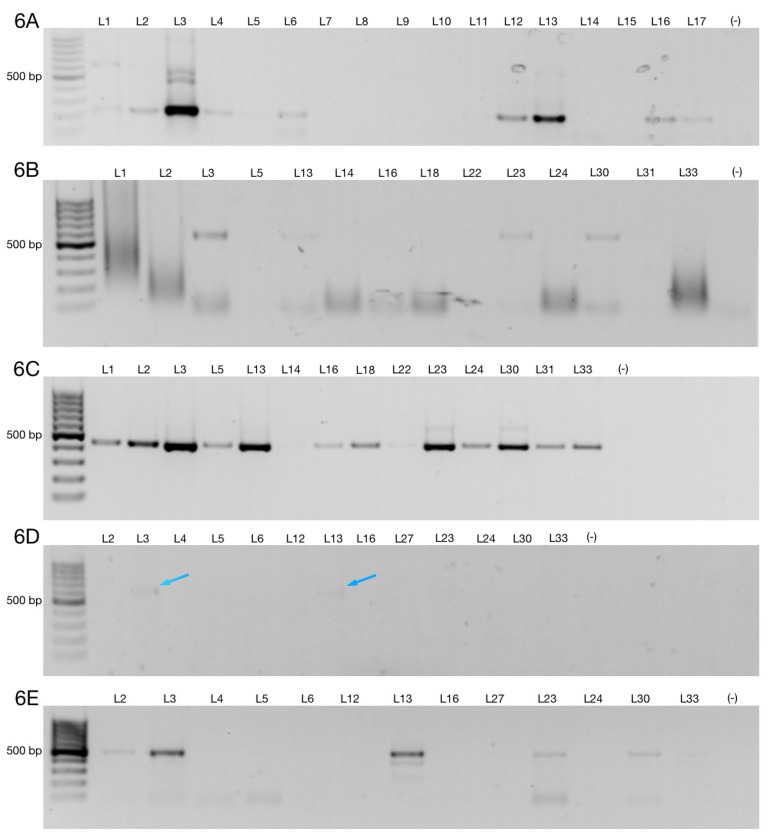
ZNF695 transcript variants are expressed in B-ALL patients. (**A**) Agarose gel showing the analysis of a representative sample expressing ZNF695_TV3. (**B**) Agarose gel showing samples with very low but positive expression of ZNF695_TV4. (**C**) Agarose gel showing samples positive for ZNF695_TV5 expression. (**D**) Expression of ZNF695_TV6 in B-ALL. We observed low expression of ZNF695_TV6 in some samples, as indicated by the blue arrows. (**E**) Expression of ZNF695_TV7 in B-ALL.

**Figure 7 genes-10-00716-f007:**
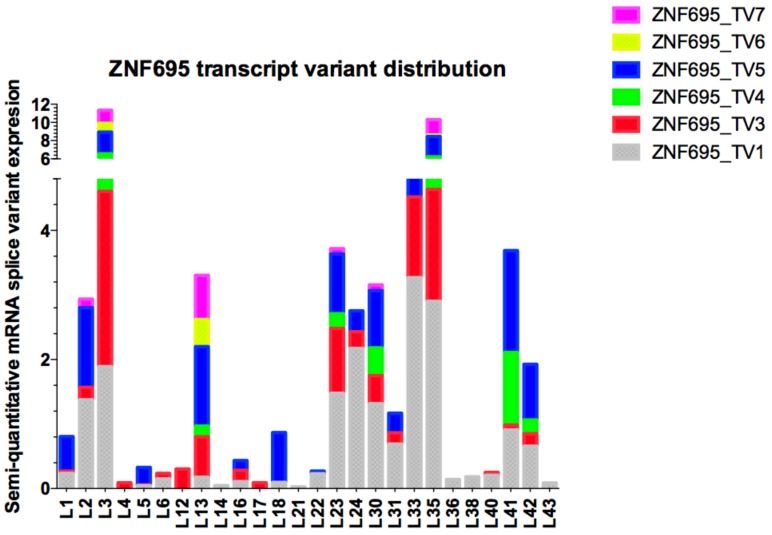
Semi-quantitative expression analysis of ZNF695 transcript variants expressed in B-ALL. The plot shows the diversity in the expression and co-expression of ZNF695 alternative splice variants in B-ALL. The color bars represent the ZNF695 transcript variants: gray bars indicate ZNF695_TV1, red bars indicate ZNF695_TV3, green bars indicate ZNF695_TV4, blue bars indicate ZNF695_TV5, yellow bars indicate ZNF695_TV6, and pink bars indicate ZNF695_TV7 expression.

**Figure 8 genes-10-00716-f008:**
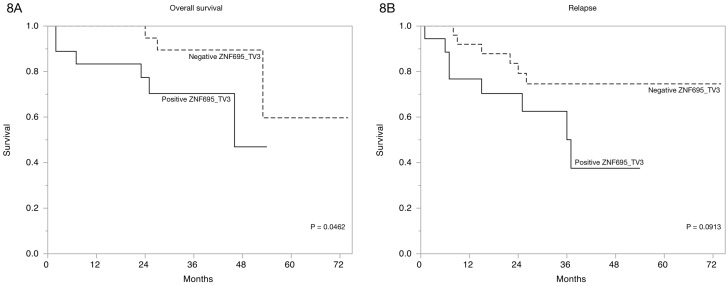
Expression of ZNF695_TV3 is associated with overall survival in B-ALL. (**A**) Kaplan-Meier survival curve indicating the association between survival and positive ZNF695_TV3 expression. (**B**) Plot indicating the association between relapse tendency and ZNF695_TV3 expression.

**Table 1 genes-10-00716-t001:** Primers designed to amplify the zinc finger 695 alternative splice variants.

Transcript	Primer Sequence	T_M_	Amplicon Size
RPL4	Forward: 5′ CGAATGAGAGCTGGCAAAGGCAAA 3′		
	Reverse: 5′ ACGCCAAGTGCCGTACAATTCATC 3′	60 °C	243 bp
ZNF695_TV1	Forward: 5′ CTGGAGAGGCTCTTTGTACTTG 3′		
	Reverse: 5′ GATAGGTTAACGTTGGTGGTAGG 3′	57.2 °C	279 bp
ZNF695 TV2/TV3	Forward: 5′CCTTTGCCTTCTGCCATGAT 3′		
	Reverse: 5′ TTAATTCAGAACTCGGGCTGAC 3′	55.2 °C	212 bp
ZNF695_TV4	Forward: 5′ GTGGCCTGCAGGGACTATTG 3′		
	Reverse: 5′ TGCAAGAGACATTGCCACATTC 3′	56.5 °C	591 bp
ZNF695_TV5	Forward: 5′ CTCCCTTGGTCTTGCTATGT 3′		
	Reverse: 5′ TGCAAGAGACATTGCCACATTC 3′	54.5 °C	425 bp
ZNF695_TV6	Forward: 5′ GTGGCCTGCAGGGACTATTG 3′		
	Reverse: 5′ TTAATTCAGAACTCGGGCTGAC 3′	56.5 °C	638 bp
ZNF695_TV7	Forward: 5′ CTCCCTTGGTCTTGCTATGT 3′		
	Reverse: 5′ TTAATTCAGAACTCGGGCTGAC 3′	54.5 °C	472 bp

**Table 2 genes-10-00716-t002:** Distribution of ZNF695 transcript variant expression in B-cell acute lymphoblastic leukemia (B-ALL) patients according to clinicopathological variables.

		ZNF695 Splice Variant Expression					
Age	Total Patients	ZNF695_TV1	ZNF695_TV3	ZNF695_TV4	ZNF695_TV5	ZNF695_TV6	ZNF695_TV7
<10 years	14 (32.5%)	6 (40%)	4 (37.7%)	4 (37.7%)	3 (20%)	2 (13.4%)	2 (13.4%)
>10 years	29 (67.5%)	16 (57.2%)	14 (50%)	12 (42.9%)	4 (14.3%)	1 (3.6%)	4 (14.3%)
**Sex**							
Male	22 (51.1%)	8 (40%)	7 (35%)	5 (25%)	1 (5%)	1 (5%)	2 (10%)
Female	21 (48.9%)	14 (60.9%)	11 (38%)	11 (38%)	6 (20.7%)	2 (6.9%)	4 (13.8%)
**Leukocytes**							
<50,000	32(74.4%)	17 (53.2%)	15 (46.8%)	11 (34.4%)	4 (15.7%)	3 (9.4%)	4 (15.7%)
>50,000	11 (25.6%)	5 (45. 6%)	3 (27.3%)	5 (45. 6%)	2 (18.2%)	0 (0%)	1 (9.1%)
**Hemoglobin**							
<10 g/dL	33 (76.7%)	17 (51.6%)	14 (42.5%)	13 (39.4%)	7 (21.3%)	3 (9.1%)	5 (18.2%)
> or = 10 g/dL	10 (23.3%)	5 (50%)	4 (40%)	3 (30%)	0 (0%)	0 (0%)	1 (10%)
**Hypodiploidy**							
Positive	10 (23.2%)	4 (40%)	4 (40%)	5 (50%)	2 (20%)	0 (0%)	1 (10%)
Negative	33 (76.8%)	18 (54.6%)	14 (42.5%)	12 (36.4%)	5 (15.2%)	3 (9.1%)	5 (15.2%)
**Philadelphia chromosome**							
Positive	2 (4.6%)	2 (100%)	2 (100%)	2 (100%)	1 (50%)	0 (0%)	1 (50%)
Negative	41 (95.4%)	20 (48.8%)	16 (39.1%)	14 (34.2%)	6 (14.6%)	3 (7.4%)	5 (12.2%)
**Relapse**							
Positive	15 (34.8%)	8 (53.4%)	8 (53.4%)	7 (46.6%)	2 (13.4%)	1 (6.7%)	3 (20%)
Negative	27 (65.2%)	14 (51.9%)	11 (40.8%)	9 (33.4%)	5 (18.6%)	2 (7.5%)	3 (11.2%)
**Survival**							
Positive	34 (79%)	15 (44.9%)	12 (35.3%)	11 (32.4%)	6 (17.7%)	3 (8.9%)	5 (14.7%)
Negative	9 (21%)	7 (77.8%)	6(66.7%)	5 (44.5%)	1 (11.2%)	0 (0%)	1 (11.2%)

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
