# Peer review of "Expression of ZNF695 Transcript Variants in Childhood B-Cell Acute Lymphoblastic Leukemia"

_genes, 2019, doi:10.3390/genes10090716_

Round 1

Reviewer 1 Report

The study by Ricardo de la Rosa is focused on ZNF695 splicing variants in four cancer cell lines and B-ALL samples. It contains interesting observations, especially those related to patient prognosis and variant expression.  The study uses a strategy of appropriate primer design for scrupulous investigation of variant content in tested material. In general, the manuscript is well written, has detailed method description and the workflow was well design. It is mainly analytical study. The major issue is a limited number of the techniques used. Moreover, the variant description part is too vaguely described, thus it is difficult to navigate in the text at the beginning of Results section. There are certain issues that should be addressed to improved readability and clarity of the text before the publication.

Major issues

1.       For better clarity and readability, the authors should use Figure 4 as Figure 1. The schema should also include the arrows indicating the position of each primer pair used in the analysis, length of a given PCR product and the nucleotide number for each exon start and end, nucleotide numbering for the splicing signals, and exon length. Also, the authors should include at the beginning of the Result section some brief introduction into ZNF695 gene structure, differences between splice variants and splicing mechanism, if possible. In both cases, RefSeq /ensemble IDs should be provided if known.

2.       Lines 239-244: it is confusing. The authors state the ZNF695_v4 and V5 variants may have alternative 3’ site, but the sequence has no identity of ZNF695? Which sequence has no identity of ZNF695? Please specify.  Is it V4 and V5? How can we be sure that 2A and 2B images sow ZNf695 variant and not some artifact? Also, the sequencing data is not shown. The authors should attempt to BLAST the sequence to identify it. May it be 670-695 read-through lncRNA? The “believing” and getting conclusions based not on experimental data, but “previous PCR design” is not convincing. What does “previous” refer to?

Minor issues

3.       1st page 40-41st line: survival rates are different than in the abstract. This is confusing.

4.       Page 2, 51st line – there is much more zinc finger-containing proteins than 100 in mammals. In human KRAB-ZNF factors alone constitute a family of more than 380 genes. Please clarify this issue.

5.       Table 2 – please indicate what the number in parentheses means.

6.       Line 189 and Line 191 – please indicate the variants that were amplified (as shown in fig 1b), provide the expected sizes and reference for “previously observed”.  It seems from the gel that the variant composition between the cell lines differs slightly.  Also, provide a better legend for the marker bands. Why only TV1, TV2, and TV3 were amplified (fig. 1 c, 1d)? Which primers were used? This part needs a better explanation.

7.       Line 191 the phrase “however, low expression in the four cell lines was determined” has strange construction and should be re-written.

8.       For better clarity, please provide a better legend for the marker bands. Also, it would be easier to read the images if the authors provided the names and expected sizes of specific gene/variant presented at each gel. Moreover, please provide the variants’ names corresponding to specific bands on the gel.  

9.       Fig 2A caption indicates co-expression of TV4 and TV5 variants in four cell lines, however in RS4 and SUP-B15 the variant composition is different than MCF7 and HeLa. Please clarify this issue.

10.   Fig 2C and 2D show pairs of bands of different sizes, although the authors refer to one variant. Please explain the visible difference.

11.   It is unclear how the primers used in fig 2B discriminate between V4 and V5. The same problem is with fig 2D and V6 and V7.

12.   Lines 206-207 – the authors should explain in more details how they chose the sites for primers and amplify unknown ZNF695 variants.

13.   Please increase the font size on your images.

14.   Figure 3b, line 250 – please add the word “respectively” after ZNF695_TV7.

15.   Figure 4 – what is the difference between gray and blue boxes?

16.   Lines: 263-264, 275-276: a very faint band for the V1 amplicon is visible. Although it is clearly very low expression, the authors should avoid a statement that there is no expression.

17.   The authors should also provide an RT-PCR image for V3 in control tissue samples and comment it in the text.

18.   Line 269 and Figure 7 have mixed information. Please correct. Also, it would be clearer to write that: different samples expressed either TV3 or TV1.

19.   Lines 311-313: please be more specific in describing the data – what is the nature of this association (e.g., better survival in patients negative for TV3).

20.   Line 348-349: certain analyses of the ZNF695 isoform content in various cancer types is presented in the study by Machnik et al., 2019, PMID  30444046. Nevertheless, this data was based on known transcript variants. 

21.   Line 371: probably dot missing and unnecessary “as”.

22.   Line 377: Please note that the ChIP-exo experiment for ZNF695 full-length protein was performed within the scope of the study by Imbeault M et al., 2017 PMID:  28273063. The raw and initially processed data is available online.

Author Response

Major issues

1.       For better clarity and readability, the authors should use Figure 4 as Figure 1. The schema should also include the arrows indicating the position of each primer pair used in the analysis, length of a given PCR product and the nucleotide number for each exon start and end, nucleotide numbering for the splicing signals, and exon length. Also, the authors should include at the beginning of the Result section some brief introduction into ZNF695 gene structure, differences between splice variants and splicing mechanism, if possible. In both cases, RefSeq /ensemble IDs should be provided if known.

We consider that is not appropriated use Figure 4 as Figure 1, because, the figure 4 is the summarized data of the ZNF695 alternative splicing our findings in the cell lines. We include the arrows indicating the position of each primers pair used for each transcript variant ZNF (variant 1, NM_020394.4 in color blue and transcript variant 3, NR_037892 in color green) and size of the PCR products are shown in the figure 2 and table 1.

Respect to the nucleotide number for each exon start and end, these data is partially reported previously study Juarez-Mendez, S. doi:10.1186/1757-2215-6-61. We previously indicated two alternative splicing sites with covering from exon 1 to 3. For this reason, in this study we design selective reverse primer that annealing in the 3 ending of the transcript variant 1 and 3 as shown in the figure 2.

We include a brief introduction in the results section that is the following: The ZNF695 gene, encode to ZNF-KRAB protein base on their protein domains suggesting a negative regulation. However, there are few studies that focused in the functions of alternative mRNA spliced. Moreover, there are not studies that show the mechanism of alternative splicing of the ZNF695. 

2.       Lines 239-244: it is confusing. The authors state the ZNF695_v4 and V5 variants may have alternative 3’ site, but the sequence has no identity of ZNF695? Which sequence has no identity of ZNF695? Please specify.  Is it V4 and V5? How can we be sure that 2A and 2B images sow ZNf695 variant and not some artifact? Also, the sequencing data is not shown. The authors should attempt to BLAST the sequence to identify it.May it be 670-695 read-through lncRNA? The “believing” and getting conclusions based not on experimental data, but “previous PCR design” is not convincing. What does “previous” refer to?

We identify the identity of the ZNF695 transcript variants TV1, TV3 (figure 1C and 1D, respectively) TV4, TV5, TV6 and TV7 (figure 2A- 2D) and we represented the most important sequences in the figure 2, because these sequences represent the identity of theses transcript variants. However, the full length of the TV4, TV5, TV6 and TV7 is unknown. In order to elucidate the full sequence of these transcripts we employed 3 RACE system. Nevertheless, the size of the TV4, TV5 expected were more to 3000 bp. We concluded in the figure 3 that the 3 RACE is no appropriated method to identify the full length of TV4, TV5, 

We modified the sentence as follows: However, the amplicons of ZNF695_TV4 and ZNF695_TV5 not were found according to the size (Figure 3B). Additionally, the sequence of these amplicons no were characterized because several bands were observed, resulting in the overlapping sequences. Although we did not obtain the full-length sequences of the ZNF695 transcript variants TV4, TV5, TV6 and TV7 using 3´RACE. However, our previous sequences Figure 2E-H showed that the ZNF695 gene could generate seven alternative transcripts, of these only six were expressed in the tested cancer cell lines as shown in the model of AS of ZNF695 in Figure 4.

Minor issues

3.       1st page 40-41st line: survival rates are different than in the abstract. This is confusing. 

The sentence was corrected

4.       Page 2, 51st line – there is much more zinc finger-containing proteins than 100 in mammals. In human KRAB-ZNF factors alone constitute a family of more than 380 genes. Please clarify this issue.

The sentence was modified as follows. The zinc finger (ZNF) protein family is the largest family of DNA-binding proteins in mammals, the ZNF proteins showed a large number of motifs that include the Cys2-His2, GATA, RanBP, A20, LIM, MYND, RING, PHD and TAZ; of these the most common are Cys2-His2 domain-containing ZNF proteins [4,5]. On the other hand, another conserved domain present in one-third of all ZNF proteins [7]is the Krüppel-associated box (KRAB) [6]. 

5.       Table 2 – please indicate what the number in parentheses means.

The number indicate the percent of positive samples that were expressed ZNF695 transcript variants. 

6.       Line 189 and Line 191 – please indicate the variants that were amplified (as shown in fig 1b), provide the expected sizes and reference for “previously observed”.  It seems from the gel that the variant composition between the cell lines differs slightly.  Also, provide a better legend for the marker bands. Why only TV1, TV2, and TV3 were amplified (fig. 1 c, 1d)? Which primers were used? This part needs a better explanation.

Thanks for your comments, we added the information as follows: three amplicons of the 400 bp, 360 pb and 310 bp previously reported were found (Figure 1B). 

We think that the differences of the ZNF695 expression among four cell lines, probability is by origin tissues because that include breast, cervical and adults and childhood leukemia (RS4 and SUP-B15, respectively).

We include more details and explanations in the figures C and D as follows:C) ZNF695_TV1 expression in the four cell lines. We design specific primers in the 3’ ending of the ZNF695 transcript variant 1 and size of 279 bp (table 1). The SUP-B15 cell line exhibited the highest expression. D) We design specific primers in the 3’ ending of the ZNF695 transcript variant 2 or 3 and size of 212 bp (table 1). ZNF695_TV2/3 was expressed in two cell lines (MCF-7 and SUP-B15) and the thinly expression in HeLa and RS4.

7.       Line 191 the phrase “however, low expression in the four cell lines was determined” has strange construction and should be re-written.

The sentence is confusing, we exclude “however, low expression in the four cell lines was determined”, because no provided information.

8.       For better clarity, please provide a better legend for the marker bands. Also, it would be easier to read the images if the authors provided the names and expected sizes of specific gene/variant presented at each gel. Moreover, please provide the variants’ names corresponding to specific bands on the gel.

In the figure 1 was include the size of the bands, the site of the amplification in the both transcripts. is important indicate that is not possible design specific primers for ZNF695_TV2 and 3, because only four nucleotides differentiate them.

9.       Fig 2A caption indicates co-expression of TV4 and TV5 variants in four cell lines, however in RS4 and SUP-B15 the variant composition is different than MCF7 and HeLa. Please clarify this issue.

In fact, RS4 and SUP-B15 are linfoblastic cell line, RS4 of adult leukemia and SUB-B15 is pediatric leukemia. The cell remains (HeLa and MCF-7) were used because in our previously report in both cell lines was identified partial transcript of variant V4 and V5. Additionally, our primers design to amplified the TV4 and TV5 are not mutually exclusively, it is not possible amplified the TV4 all time is coexpressed TV5 basically by the primers to binding SS1 site is before of the SS2. Moreover, is the unique chance to amplified both transcript because the primers binding in the alternative site. We think that the composition of the expression is different in all cell lines inclusively in the MCF7 and HeLa, because the bands are expressed in different intensity as well as RS4 and SUP-B15. 

10.   Fig 2C and 2D show pairs of bands of different sizes, although the authors refer to one variant. Please explain the visible difference.

Thanks for your comments, we observed different size between 2c and 2D by in both we use different forward primer SS1 and SS2 for 2C and 2D, respectively. In the figure 2C, is observed a slight difference between MCF-7 and HeLa cell lines, therefore, both amplification products were sequenced and the same sequence was obtained, so the slight difference in weight between the was probability due to the effect of the intercalant Sybr gold used in the electrophoresis. Moreover, the sequence showed the identity of the alternative transcripts 2E-2H. The great challenge is determinate the abundance of each transcript and the full length. We are working today in NGS using the Pacbio system to inspection the complete sequence of the ZNF695 transcript variants.

In the figure 2 we include the length of the PCR products 2A-2D. 

11.   It is unclear how the primers used in fig 2B discriminate between V4 and V5. The same problem is with fig 2D and V6 and V7.

Is not possible discriminate between V4 and V5, because the V5 transcript composition showed two alternative sites Figure 4. However, V5 is more expressed than V4 according to 2A y 2B. In the case of V6 and V7 is the same issue, because both of them transcripts showed the same alternative sites. However, here the results are inverses to those found in the figure 2A-B. The V6 is more expressed than V7, additionally only HeLa and MCF-7 expressed these transcripts.

It is important to mention that the V4, V5, V6 and V7 transcript share the same alternative sites of splicing, however the primer reverse give the identity to differentiate among of them.

12.   Lines 206-207 – the authors should explain in more details how they chose the sites for primers and amplify unknown ZNF695 variants.

As specified in Table 1, these primers were designed to selectively amplify the uncharacterized ZNF695 splice variants when combined with the amplification mixture. We design two forward primers that are complement to SS1 and SS2, the reverse primers were specific to 3’ ending of the ZNF695_TV1, NM_020394.4, these amplification mixtures amplified V4 and V5. Additionally, we used the same forward primers SS1 and SS2 the reverse primers were specific to 3’ ending of the ZNF695_TV2, NM_001204221.1 or ZNF695_TV3, NR_037892.1, long noncoding RNA, these amplification mixtures amplified V6 and V7.

13.   Please increase the font size on your images.

We increased the font size of the figures.

14.   Figure 3b, line 250 – please add the word “respectively” after ZNF695_TV7.

The word was included.

15.   Figure 4 – what is the difference between gray and blue boxes?

We modified the color box, the grey box indicates the alternative spliced site 1 (SS1) and the blue box indicate the alternative spliced site 1 (SS2) in different ZNF695 transcript variants. In the figure legend was included.

16.   Lines: 263-264, 275-276: a very faint band for the V1 amplicon is visible. Although it is clearly very low expression, the authors should avoid a statement that there is no expression.

Thanks for the comment, we modified the sentence as follows: Interestingly, we observe a thinly amplification in the healthy controls 

17.   The authors should also provide an RT-PCR image for V3 in control tissue samples and comment it in the text.

We consider that is not necessary included the figure of the V3, because was not expressed in the control samples, as follows in the figure V3, but if is necessary we should include the correspond figure. 

               C1        C2        C3       C4       C5     C6       C7      C8        C9      RS4 SUP-B15  (-)

Figure V3. 

18.   Line 269 and Figure 7 have mixed information. Please correct. Also, it would be clearer to write that: different samples expressed either TV3 or TV1.

We modified the sentence. Moreover, we identify that the expression of ZNF695_TV3 and ZNF695_TV1 are mutually exclusive because are expressed in different samples as follows: ZNF695_TV3 (samples 8, 12 16 and 17) and ZNF695_TV1 (samples 4, 5, 10 and 14) (Figure 7).

19.   Lines 311-313: please be more specific in describing the data – what is the nature of this association (e.g., better survival in patients negative for TV3).

We included more details as follows: Significant association between overall survival and relapse and ZNF695_TV1 expression (data not shown) was not observed. However, the Kaplan-Meier curves based on ZNF695_TV3 expression revealed statistically significant differences in the overall survival and the tendency to relapse (Figure 8). These results are shown that the expression of ZNF695_TV3 is correlation with poor survival.

20.   Line 348-349: certain analyses of the ZNF695 isoform content in various cancer types is presented in the study by Machnik et al., 2019, PMID  30444046. Nevertheless, this data was based on known transcript variants. 

We modified the sentence as fallowing: no previous studies have identified and quantified the expression of the novel ZNF695 transcript variants in leukemia childhood

21.   Line 371: probably dot missing and unnecessary “as”.

Thanks for the comment, we delete as 

22.   Line 377: Please note that the ChIP-exo experiment for ZNF695 full-length protein was performed within the scope of the study by Imbeault M et al., 2017 PMID:  28273063. The raw and initially processed data is available online.

Thanks very much for you comment, one of the objectives in the laboratory is to know the binding sites of the ZNF695_TV1, TV4 and TV5. We are going to consider for future studies the data of the study by Imbeault M. 

Reviewer 2 Report

In this paper, de la Rosa and colleagues investigated the expression patterns of six ZNF695 transcript variants in various cancer cell lines, in a cohort of patients with childhood B-cell acute lymphoblastic leukemia (B-ALL), and finally report correlation between noncoding lncRNA ZNF695_TV3 expression and survival of patients in childhood B-ALL.

The study is built on the previous observation of the research group where they identified new splice variants of ZNF695 in ovarian cancer. In this study, the authors report coexpression of six     alternative transcripts of the ZNF695 gene in five different cell lines of different tissue origin (cervical cancer, breast cancer, leukemia).

Although the paper is well written and carries data with potential novelty and of clinical interest, several shortcoming hamper the enthusiasm of accepting the paper for publication in its present form.

Major comments:

- Although the title of the paper suggests relevance of ZNF695 expression in childhood B-ALL, the authors spend a significant proportion of the manuscript on reporting the detailed results of the ZNF695 expression in the five cell lines analysed, only one of which is a leukemia cell line. This should be more balanced with a focus on the ALL part of the data and more detailed presentation of this part of the study.

- Along these lines, practically there is no clinical data about the pediatric ALL cohort the authors used. Basic clinical and molecular (e.g. cytogenetic alterations) features would be essential to include in order to gain better understanding between the clinical/pathological features of patients and ZNF695 expression.

-The detailed ZNF695 transcript variant expression data is summarized only on Figure 7 with semiquantitative data obtained from densitometric analysis of the agarose gels. Could this be better quantified using real-time PCR assays for the most relevant transcript variants?

- This reviewer wonders whether a simple PCR approach followed by a semiquantitative assessment of the agarose gel images is a reliable way of differentiating between the transcript variants. Given the potential novelty of these observations, at least the major transcript variants should be analysed and quantified using quantitative real-time PCR to confirm and extend the findings of the standard PCR assays.

Minor comments:

- The description of the methods is very long with all time and temperature details reported accurately. Unless, this is a specific requirement from the journal, I think this could be shortened significantly.

- The authors state that the B-ALL cases negative for ZNF695 expression are likely methylated. This should be confirmed experimentally to demonstrate differential methylation between the ZNF695 expressors and cases negative for ZNF695 expression.

Author Response

Major comments:

- Although the title of the paper suggests relevance of ZNF695 expression in childhood B-ALL, the authors spend a significant proportion of the manuscript on reporting the detailed results of the ZNF695 expression in the five cell lines analysed, only one of which is a leukemia cell line. This should be more balanced with a focus on the ALL part of the data and more detailed presentation of this part of the study.

We previously identify two alternative splicing sites with covering from exon 1 to 3 Juarez-Mendez, S. doi:10.1186/1757-2215-6-61. Additionally, observed the expression of these variants in 10 different cancer cell lines, that include leukemia. Additionally, ZNF695 has been associated to prognosis in several malignances. However, the transcript variants of ZNF695 has not been associated to the prognosis. For this reason, in this study we design selective reverse primer that annealing in the 3 ending of the transcript variant 1 and 3 as shown in the figure 2. In order to know the diversity of the transcript variant expressed in cancer cell lines and subsequently in leukemia patients. 

- Along these lines, practically there is no clinical data about the pediatric ALL cohort the authors used. Basic clinical and molecular (e.g. cytogenetic alterations) features would be essential to include in order to gain better understanding between the clinical/pathological features of patients and ZNF695 expression.

We had a diverse clinical data as leucocyte count, cytogenetic alterations, treatment response, survival, among others. However, these data do not provide relevant information to this work. We performed several analyses but not observed a significant correlation. We only determinate the overexpression of variant ZNF695_V3 had a statistically significant correlation, probability by the number of patients include in this study. In the futures studies we are considering to increase the number of patients

-The detailed ZNF695 transcript variant expression data is summarized only on Figure 7 with semiquantitative data obtained from densitometric analysis of the agarose gels. Could this be better quantified using real-time PCR assays for the most relevant transcript variants?

The approach for the identification of new variants of alternative splicing is necessary to use a laborious and scrupulous strategy to know in a specific each variant in this case ZNF695 splicing.  For this, it was necessary to design several primers pair that amplifies a single variant and same time discriminate the others. This design to variants splicing of ZNF695 is complex but principal size is big, between ~200-600 nt, for this reason is impossible quantified using real-time PCR, due to the small amplicon size of qPCR assays helps to minimize assay-related problems, according to Bustin SA et al., PMID: 19246619. 

- This reviewer wonders whether a simple PCR approach followed by a semiquantitative assessment of the agarose gel images is a reliable way of differentiating between the transcript variants. Given the potential novelty of these observations, at least the major transcript variants should be analysed and quantified using quantitative real-time PCR to confirm and extend the findings of the standard PCR assays.

The semiquantitative analysis for gene expression are very common method used when the transcript are long to be quantified by real time PCR. 

Minor comments:

- The description of the methods is very long with all time and temperature details reported accurately. Unless, this is a specific requirement from the journal, I think this could be shortened significantly.

We shorten and improve the methods according to the recommendations of reviewers. 

- The authors state that the B-ALL cases negative for ZNF695 expression are likely methylated. This should be confirmed experimentally to demonstrate differential methylation between the ZNF695 expressors and cases negative for ZNF695 expression.

In fact, previous study according to Takahashi T et al, PMID: 25273507 showed that ZNF695 non methylation were significantly associated with poor chemoradioterapy response in esophageal squamous cell carcinoma, these results are sugguesting that pacients non-responder have ZNF695 overexpression. For this reason, we suggest in discussion that potential methylated in sample of LLA without ad ZNF695, but we need to confirm this hypothesis in futures studies.

Round 2

Reviewer 1 Report

In general, the answers to my questions and the new text inserted in the revised manuscript harbor many grammar mistakes. Sometimes, it is very difficult to understand the authors. Please, have the text and answers corrected by a professional language editor before re-submission.

1.       The information about the known ZNF695 splicing variants is missing. Please provide at the beginning of the Result section (or in the Introduction) information on the ZNF695 gene structure – not only main protein domains, but also the number of exons and introns in the longest variant. Also, please provide a list of annotated splicing variants with their RefSeq or Ensembl ID. It would be helpful if the authors briefly summarize how these annotated splicing variants differ from the longest variant. 

2.       Figure 3B – the image lacks red and yellow arrows, they may remain (as it was in v1) for clarity. Lines 233-236 are improved, but still unclear. In the previous version the authors mentioned that the sequence wasn’t similar to ZNF695, while in the current version they state that these sequences were not characterized due to overlapping sequences. So what was the result of sequences? Did the authors receive full sequences for these unpredicted bands (~1500 nt and ~1200nt)? Or the sequences were unreadable due to overlapping signal? If the sequence was overlapping, than the authors should sequence each band individually (e.g. – cut-out from the gel and clone into a sequencing plasmid like pGEM).

3.       Line 180 – Figure 1b - a comment on the uneven band profile between the samples and  potential explanation for such observation should be added within the manuscript text.

4.       As mentioned previously, the authors should provide more information on their figures to facilitate the read-out of their data. Although all necessary information is provided in the text, caption or table, additional information (such as: marker bands size, amplicon sizes, variant names) will be advantageous. Otherwise the reader will have to jump from one part of the article to another to find the required details.

This is an example:

(see attached)

5.       In their cover letter, the authors indicated that it is impossible to design the primers individually to TV2 and TV3 variants, because they differ in four nucleotides (answer no.8). If there is four-nucleotide difference between the variants, then these nucleotides might be included (or excluded) at the 3’end of the primers in order to distinguish TV2 from TV3.  Such a test should be performed. Otherwise, please explain in more details (including the sequence of both variants) why the design of primers is not possible.

6.       Wherever there is an uneven profile of variant(s’) expression between various cell lines, the authors should highlight and explain this fact.

7.       Fig 2A – please add the length of TV4, when using this primer pair. Also, refer to comment no. 4 and provide the names of each variant in Figure 2.

8.       As far as answer 10 in the cover letter is concerned, please provide as a supplementary material the sequences of both amplicons representing TV6 for HeLa and MCF7 (fig 2C) and TV7 for Hela and MCF7 (Fig 2D). In the text please also refer to observed difference in bands’ sizes (Fig 2C & 2D) and provide the potential explanation to this observation.

9.       Figure 5C caption, lines: 271-272 – please change the caption to adhere with the changes in the text, and to pinpoint that very low expression of the TV1 variants was observed in healthy controls.

10.   Line 265- there is a mistake: TV1 should be TV3 and vice versa (according to your Figure 7 legend).

11.   Lines 308-309: Please re-write: These results showed that the expression of ZNF695_TV is assiciated with poor survival and higher tendency to relapse.

Author Response

In general, the answers to my questions and the new text inserted in the revised manuscript harbor many grammar mistakes. Sometimes, it is very difficult to understand the authors. Please, have the text and answers corrected by a professional language editor before re-submission.

Thank you for the comment, the manuscript was edited for expert, we include the certificate in the end text.

The information about the known ZNF695 splicing variants is missing. Please provide at the beginning of the Result section (or in the Introduction) information on the ZNF695 gene structure – not only main protein domains, but also the number of exons and introns in the longest variant. Also, please provide a list of annotated splicing variants with their RefSeq or Ensembl ID. It would be helpful if the authors briefly summarize how these annotated splicing variants differ from the longest variant.

The ZNF695 gene is localized on Chromosome 1 and the reverse strand. Two transcripts encode proteins. The first, the longest transcript, consists of four exons with a total transcript length of 3,341 bp. This variant is characterized by a very long exon 4 of 2,933 bp that encodes a protein with 515 aa (ENST00000339986.8) in the NCBI database (NM_020394.5, TV1) and encodes the ZNF-KRAB protein that belong to the families ZNF and Cys2-His2. The second, the short transcript, has six exons with a length of 826 bp (ENST00000487338.6) and 919 bp in NCBI (NM_001204221, TV2). However, this protein contains no ZNF domain. Additionally, the ZNF695 gene is transcribed to the ZNF695 long noncoding RNA (ENST000000498046.2, 504 bp); however, in the NCBI database, the noncoding transcript has 923 bp (NR_037892.2, TV3), which has four nucleotides more than the transcript variant 2 NM_001204221. Finally, three nonsense-mediated decay transcripts are reported in Ensembl; these transcripts comprise six exon and showed diverse sequences (ENST00000491337.6, ENST00000479214.5, ENST00000366504.6, with 714, 885 and 862 bp, respectively). Hereafter, we only used the sequences that are reported in the NCBI database.

Figure 3B – the image lacks red and yellow arrows, they may remain (as it was in v1) for clarity. Lines 233-236 are improved, but still unclear. In the previous version the authors mentioned that the sequence wasn’t similar to ZNF695, while in the current version they state that these sequences were not characterized due to overlapping sequences. So what was the result of sequences? Did the authors receive full sequences for these unpredicted bands (~1500 nt and ~1200nt)? Or the sequences were unreadable due to overlapping signal? If the sequence was overlapping, than the authors should sequence each band individually (e.g. – cut-out from the gel and clone into a sequencing plasmid like pGEM).

In order to characterize the complete sequence of the ZNF695 of the variants (TV4 and TV5), we were made RACE ´3 in cancer cell lines. We expected four amplicons: ZNF695_TV4, 3215 bp; ZNF695_TV5, 3049 bp; ZNF695_TV6, 781 bp; and ZNF695_TV7, 615 bp. However, our result was as follows: the amplicons of ZNF695_TV6 and ZNF695_TV7 were of the expected size, as shown by the green and blue arrows, respectively (Figure 3B). However, the amplicons of ZNF695_TV4 and ZNF695_TV5 not were found according to the expected size (Figure 3B). The remaining amplified were first cut, purified and sequenced, but, we did not obtain any sequences. Therefore, we cloned into pJET cloning vector and sequences, the sequence obtained were unreadable. We thought that the PCR products are unspecific. For this reason, in the first version we had a word error, so the correct word was non-characterization.

Line 180 – Figure 1b - a comment on the uneven band profile between the samples and potential explanation for such observation should be added within the manuscript text.

Thank you for the comment, we include more details in the figure legend.

Figure 1b)ZNF695 expression, including all transcript variants that were previously reported [46]. All cell lines expressed at least one of the ZNF695 transcripts, RS4 only expressed the ZNF695 variant TV1 and/or TV3 of ~400 bp, while the HeLa and SUP-B15 cell lines expressed two variants of ZNF695, which corresponded the expected lengths of 400 bp (TV1/TV3) and 360 bp (TV4/TV6). Additionally, we observed other variants of ~200 bp in these cell lines; however, we focused on the previous variants described. Finally, the MCF-7 cell lines coexpressed the three splicing variants of 400 bp (TV1/TV3), 360 bp (TV4/TV6) and 310 bp (TV5/TV7)

As mentioned previously, the authors should provide more information on their figures to facilitate the read-out of their data. Although all necessary information is provided in the text, caption or table, additional information (such as: marker bands size, amplicon sizes, variant names) will be advantageous. Otherwise the reader will have to jump from one part of the article to another to find the required details.

This is an example:

(see attached)

Thank you for your observation, the information has already been added in the figure 1 and 2 caption.

Figure 1. Alternative ZNF695 transcript variants are expressed in cancer cell lines. Gene expression was evaluated in MCF-7, HeLa, RS4 and SUP-B15 cell lines. A) We observed homogenous expression of the RPL4 housekeeping gene in the four cell lines. B) ZNF695 expression, including all transcript variants that were previously reported [46]. All cell lines expressed at least one of the ZNF695 transcripts, RS4 only expressed the ZNF695 variant TV1 and/or TV3 of ~400 bp, while the HeLa and SUP-B15 cell lines expressed two variants of ZNF695, which corresponded the expected lengths of 400 bp (TV1/TV3) and 360 bp (TV4/TV6). Additionally, we observed other variants of ~200 bp in these cell lines; however, we focused on the previous variants described. Finally, the MCF-7 cell lines coexpressed the three splicing variants of 400 bp (TV1/TV3), 360 bp (TV4/TV6) and 310 bp (TV5/TV7) [46]. C) ZNF695_TV1 expression in the four cell lines. We designed specific primers for the 3’ end of the ZNF695 transcript variant 1 and a size of 279 bp (Table 1). The SUP-B15 cell line exhibited the highest expression. D) We designed specific primers for the 3’ end of the ZNF695 transcript variant 2 or 3 and a size of 212 bp (Table 1). ZNF695_TV2/3 was expressed in two cell lines (MCF-7 and SUP-B15) and barely expressed in HeLa and RS4 cells.

Figure 2. Selective amplification of ZNF695 alternative transcript variants.We selectively amplified the transcript variants of ZNF695. The black arrow and the black line indicate the sites of amplification and the PCR product size, respectively. The blue box indicates the full length of the long transcript of ZNF695 (TV1), the green box represents the short transcript of ZNF695 (TV3) and the grey box indicates the spliced region in the transcript. A) Amplification of ZNF695_TV4 (591 bp) and ZNF695_TV5 (561 bp), which were coexpressed in the MCF-7 and HeLa cell lines. We observed that RS4 and SUP-B15 only expressed one variant, TV5 and TV4, respectively. B) Specific amplification of ZNF695_TV5 of 425 bp in the four cell lines. C) Amplification of ZNF695_TV6 (638 bp) in the MCF-7 and HeLa cell lines apparently expressed a preference for a TV7 of 608 pb. D) Expression of ZNF695_TV7 (472 bp) was low in the MCF-7 and HeLa cell lines. We observed that in the HeLa cell line results was slight heavier than that of MCF-7; however, the sequence no showed differences. E) Sequence of SS1 of the amplified ZNF695 transcript variants. The blue boxes show the boundary of the AS site. F) Sequence that confirms the identity of ZNF695_TV3. G) Partial sequence that confirms the identity of ZNF695_TV1 (NM_020394.5). H) Partial sequence that confirms the identity of ZNF695_TV3 (NR_037892.1).

In their cover letter, the authors indicated that it is impossible to design the primers individually to TV2 and TV3 variants, because they differ in four nucleotides (answer no.8). If there is four-nucleotide difference between the variants, then these nucleotides might be included (or excluded) at the 3’end of the primers in order to distinguish TV2 from TV3.  Such a test should be performed. Otherwise, please explain in more details (including the sequence of both variants) why the design of primers is not possible.

Thanks for your comment. We include more information about of the sequences of TV2 and TV3 in the first section of the results. Additionally, the ZNF695 gene transcribes to the ZNF695 long non-coding RNA (ENST000000498046.2, with 504 bp of length), however in the NCBI database the non-coding transcript have a 923 bp (NR_037892.2, TV3) which one has four nucleotides more than the transcript variant 2 NM_001204221. Thus, is not possible design specific primer for amplified selectively TV2 and TV3.

Wherever there is an uneven profile of variant(s’) expression between various cell lines, the authors should highlight and explain this fact.

Thank you for your observation. In fact, in the figure 1b has already been added in the figure legend.

Fig 2A – please add the length of TV4, when using this primer pair. Also, refer to comment no. 4 and provide the names of each variant in Figure 2.

Thank you, in figure 2, in addition to the expected length, we already include in the legend figure the length of the different amplification product obtained.

As far as answer 10 in the cover letter is concerned, please provide as a supplementary material the sequences of both amplicons representing TV6 for HeLa and MCF7 (fig 2C) and TV7 for Hela and MCF7 (Fig 2D). In the text please also refer to observed difference in bands’ sizes (Fig 2C & 2D) and provide the potential explanation to this observation.

Thanks for your comment, we included the Figure S1 and S2 of the representative sequences of TV6 and TV7, respectively. The differences between TV6 and TV7 is in the Figure S2, in these sequences are showed up stream to splicing site and correspond to exon 2.

Figure 5C caption, lines: 271-272 – please change the caption to adhere with the changes in the text, and to pinpoint that very low expression of the TV1 variants was observed in healthy controls.

Thank you, we including in the text your comment.

Line 265- there is a mistake: TV1 should be TV3 and vice versa (according to your Figure 7 legend).

Thanks for your observation, we include more explication in the manuscript. In the figure 7 showed the semiquantitative expression of the six alternative transcript variants, and these transcripts were evaluated independently using the primers are shown in the table 1. Only TV2/TV3 could be amplified with the same primers, because the difference between the two variants is four nucleotides.

Lines 308-309: Please re-write: These results showed that the expression of ZNF695_TV is associated with poor survival and higher tendency to relapse.

Thanks for your observation, we modified the manuscript.

Reviewer 2 Report

The authors responded to the majority of the criticism raised in the first round of the review process, however some concerns still remain with regards to the ability of the employed methodology to clearly distinguish between the different transcript variants presented in the paper (also discussed by the other reviewer in detail).

Since the association with overall survival barely reached the level of significance, it would be necessary to state in the conclusion part that these findings are preliminary and require further validation on larger, independent cohorts. Also, it would be more appropriate to change the title of the paper referring to the expression profile of ZNF695 instead of referring to the weak association with overall survival.

Author Response

The authors responded to the majority of the criticism raised in the first round of the review process, however some concerns still remain with regards to the ability of the employed methodology to clearly distinguish between the different transcript variants presented in the paper (also discussed by the other reviewer in detail).

Since the association with overall survival barely reached the level of significance, it would be necessary to state in the conclusion part that these findings are preliminary and require further validation on larger, independent cohorts. Also, it would be more appropriate to change the title of the paper referring to the expression profile of ZNF695 instead of referring to the weak association with overall survival.

Thanks for your observation, we modified the manuscript, we include more details of our findings and modified the title and the conclusion.

The ZNF695 gene is transcribed as six alternative transcript variants in human cancer cell lines. These transcript variants were evaluated in B-ALL patients, and positive expression was found in 60.4% of the patients. We found that lncRNA ZNF695_TV3 expression was associated with poor survival and an increased tendency to relapse in patients with B-ALL. These findings are preliminary and require further validation in a large cohort.

Round 3

Reviewer 1 Report

Many thanks for the revised version of the paper.
I think if has improved and warrants publication in Genes now.

Reviewer 2 Report

Major comment:

Line 611: there is still a mistake. The authors mentioned that the single expression of TV3 is observed in samples 8, 12, 16, and 17, and TV1 in samples 4, 5, 10,” and 14. In contrast, Figure 7 shows the opposite: red bar (indicating TV3) is visible for samples 4, 5, 10 and 14, while grey bar (indicating TV1) is visible for samples 8, 12, 16, 17. Besides, the picture gets even more complicated when compared with the Figures 5 and 6. Does the sample numbering In Figure 5 and 6 correspond to the numbers in Figure 7? If so, then the interpretation of the gel images harbors many mistakes (e.g., there are variants TV1, TV3 and TV5 expressed in the sample L5). Please correct or explain the visible differences. To avoid confusion, the numbering should be unified. Also in line 610, it would be more appropriate to use the word “may be” mutually exclusive, rather than “are”, because in some samples both variants are expressed.

Minor comments:

I propose to make the following changes at the beginning of your Results (lines 246-248)

"This variant is characterized by a very long exon 4 of 2,933 bp that and encodes a protein with 515 aa in before Ensembl database (ENST00000339986.8) and the NCBI database (NM_020394.5, TV1). This variant and encodes the ZNF-KRAB protein that belongs to the families ZNF and Cys2-His2 families."

Line 256: and show

Figure caption 2C: Similarly to the caption 2D, please provide a comment about uneven band sizes.

Line 489: I suggest to re-write the phrase into: “slightly heavier; however, the sequences obtained from both cell lines showed no difference.”

Line 490: Is it Splicing Site 1 or 2? The caption mentions SS1, while there is Splicing Site 2 mentioned in the Figure2E.

Supplement caption: please indicate what exactly “four lines” mean. Do you mean the six different lines representing each exon? If so, then “four” might be changed to “the”. Also, please explain in the caption what green and blue boxes mean and indicate which sequences differentiate TV6 from TV7.

Line 94: I suggest: ovarian normal and tumor tissues

Lines 360-364 are a bit complicated. I suggest the following changes (according to my understanding) so that it is easier to understand the primer design. Please make sure that this text doesn’t contain any error related to the primer usage:

We designed two forward primers that complemented SS1 and SS2, as well as two reverse primers: one specific to the 3’ end of ZNF695_TV1 ( NM_020394.5), and the other specific to the 3’ end of ZNF695_TV2 (NM_001204221.1) or ZNF695_TV3 (NR_037892.1, a lncRNA). SS1 and ZNF695_TV1 primers amplified variants TV4 and TV5 at the same time, while SS2 and ZNF695_TV1 amplified
specifically TV5. The second reverse primer was used to amplify TV6 when paired with SS1 forward primer, or TV7 variant when paired with SS2 primer (Table 1, Figure 2 A-D).
Line 744: childhood leukemia rather than leukemia childhood
Lines 85 and 780 harbor slightly different information, please be more accurate.

Author Response

Major comment:
1.- Line 611: there is still a mistake. The authors mentioned that the single expression of TV3 is observed in samples 8, 12, 16, and 17, and TV1 in samples 4, 5, 10,” and 14. In contrast, Figure 7 shows the opposite: red bar (indicating TV3) is visible for samples 4, 5, 10 and 14, while grey bar (indicating TV1) is visible for samples 8, 12, 16, 17. Besides, the picture gets even more complicated when compared with the Figures 5 and 6. Does the sample numbering In Figure 5 and 6 correspond to the numbers in Figure 7? If so, then the interpretation of the gel images harbors many mistakes (e.g., there are variants TV1, TV3 and TV5 expressed in the sample L5). Please correct or explain the visible differences. To avoid confusion, the numbering should be unified. Also in line 610, it would be more appropriate to use the word “may be” mutually exclusive, rather than “are”, because in some samples both variants are expressed.

Thank you very much for your comments. The data in figure 7 do not correlate with the information presented in figures 5 and 6. For this reason, according to your observations we modify the image 7, order the samples and approve the information with respect to figure 5 and 6. We want also mention that we observed three data in figure 7 were inverted, which have already been modified. In addition, we observed that the numbers in Figure 6D were not aligned, however, it was already corrected.
It was modified the word “may be”, it is indeed more appropriate.

Minor comments:

1.- I propose to make the following changes at the beginning of your Results (lines 246-248) This variant is characterized by a very long exon 4 of 2,933 bp that and encodes a protein with 515 aa in before Ensembl database (ENST00000339986.8) and the NCBI database (NM_020394.5, TV1).
This variant and encodes the ZNF-KRAB protein that belongs to the families ZNF and Cys2-His2 families.

Thanks for you. We performed the changes according to the comments. (lines 168-171)

2.- Line 256: and show

We performed the changes according to the comments. (line 177)

3.- Figure caption 2C: Similarly to the caption 2D, please provide a comment about uneven band sizes.

We performed correction in the previous version. (lines 265-269)

4.- Line 489: I suggest to re-write the phrase into: “slightly heavier; however, the sequences obtained from both cell lines showed no difference.”

Thank you for the comment, we re-wrote the phrase. (line 268)

5.- Line 490: Is it Splicing Site 1 or 2? The caption mentions SS1, while there is Splicing Site 2 mentioned in the Figure2E

Thank you. We had a mistake, we corrected. (lines 269-270)

6.- Supplement caption: please indicate what exactly “four lines” mean. Do you mean the six different lines representing each exon? If so, then “four” might be changed to “the”. Also, please explain in the caption what green and blue boxes mean and indicate which sequences differentiate TV6 from TV7

Thank you for your observation, we corrected of supplementary caption. Additionally, we thinks that is not appropriated include more information in the captions, because, the strategies used for the identified the TV6 and TV7 was not allowed to obtain identify sequences of TV7 (SS1 and SS2), only partial exon 2 was sequenced through the sequencing primer used to align on SS1, that is exclusively of TV7.

7.- Line 94: I suggest: ovarian normal and tumor tissues

We performed the changes according to the comments. (line 88)

8.- Lines 360-364 are a bit complicated. I suggest the following changes (according to my understanding) so that it is easier to understand the primer design. Please make sure that this text doesn’t contain any error related to the primer usage:

We designed two forward primers that complemented SS1 and SS2, as well as two reverse primers: one specific to the 3’ end of ZNF695_TV1 ( NM_020394.5), and the other specific to the 3’ end of ZNF695_TV2 (NM_001204221.1) or ZNF695_TV3 (NR_037892.1, a lncRNA). SS1 and ZNF695_TV1 primers amplified variants TV4 and TV5 at the same time, while SS2 and ZNF695_TV1 amplified specifically TV5. The second reverse primer was used to amplify TV6 when paired with SS1 forward primer, or TV7 variant when paired with SS2 primer (Table 1, Figure 2 A-D).

We performed the changes according to the comments. (lines 222-228)

9.- Line 744: childhood leukemia rather than leukemia childhood.

We performed the changes according to the comments. (line 409)

10.- Lines 85 and 780 harbor slightly different information, please be more accurate.

We tried review the first version in the line 85 and line 780 have not relation.
Lines 84-86: However, 98% of the human transcriptome is noncoding RNA. Noncoding RNA is divided into short noncoding RNA and long noncoding RNA (lncRNA). lncRNAs are transcripts with a full length of greater than > 200 nucleotides [41]

Lines 779-781: To date, no previous studies have identified and quantified the expression of novel ZNF695 transcript variants in leukemia childhood.
Please, could you tell us the error in the current version

Round 4

Reviewer 2 Report

The text has improved. However, I have some minor suggestions to correct some language errors. Besides, I have no further comments. I recommend publication.

  1. Results, first paragraph, 3rd & 4th sentence – there was some mix-up with word editing of my suggested corrections. I suggest the following changes:
    This variant is characterized by a very long exon 4 of 2,933 bp. The variant (TV1) encodes a protein with 515 aa ZNF-KRAB protein (Ensembl database: ENST00000339986.8, NCBI database: NM_020394.5, TV1) that belongs to the ZNF and Cys2-His2 families.

    2. Figure 2D caption, 2nd sentence – there are some language mistakes in the newest version. I suggest the following changes:

    We observed that in the HeLa cell line the amplified fragment showed bigger size than MCF7. However, the sequences obtained from both fragments showed no difference.

Author Response

Dear Reviewer

I am pleased your time for reviewer our manuscript entitled “Expression of long noncoding RNA ZNF695 predicts survival in childhood B-cell acute lymphoblastic leukemia” by Ricardo de la Rosa and Sergio Juárez-Méndez for consideration for publication in the Genes. Thank you for your comments for improves it.

We have already made the correction, according to reviewer suggestions.